# Knowledge-Aware Bayesian Deep Topic Model

**Dongsheng Wang, Yishi Xu, Miaoge Li, Zhibin Duan, Chaojie Wang, Bo Chen**[*]
National Laboratory of Radar Signal Processing
Xidian University, Xi'an, Shanxi 710071, China
{wds,xuyishi,limiaoge,zhibinduan}@stu.xidian.edu.cn
xd_silly@163.com, bchen@mail.xidian.edu.cn

**Mingyuan Zhou**
McCombs School of Business
The University of Texas at Austin, Austin, TX 78712, USA
mingyuan.zhou@mccombs.utexas.edu

## Abstract

We propose a Bayesian generative model for incorporating prior domain knowledge into hierarchical topic modeling. Although embedded topic models (ETMs) and its variants have gained promising performance in text analysis, they mainly focus on mining word co-occurrence patterns, ignoring potentially easy-to-obtain prior topic hierarchies that could help enhance topic coherence. While several knowledge-based topic models have recently been proposed, they are either only applicable to shallow hierarchies or sensitive to the quality of the provided prior knowledge. To this end, we develop a novel deep ETM that jointly models the documents and the given prior knowledge by embedding the words and topics into the same space. Guided by the provided domain knowledge, the proposed model tends to discover topic hierarchies that are organized into interpretable taxonomies. Moreover, with a technique for adapting a given graph, our extended version allows the structure of the prior knowledge to be fine-tuned to match the target corpus. Extensive experiments show that our proposed model efficiently integrates the prior knowledge and improves both hierarchical topic discovery and document representation.

## 1 Introduction

Topic models (TMs) such as latent Dirichlet allocation (LDA) have enjoyed success in text mining and analysis in an unsupervised manner [4]. Typically, the goal of TMs is to infer per-document topic proportions and a set of latent topics from the target corpus using word co-occurrences within each document. The extracted topics are widely used in various machine learning tasks [20, 19, 33, 39]. However, the objective function of most TMs is to maximize the likelihood of the observed data, which causes an over-concentration on high-frequency words. Moreover, the infrequent words might be assigned to irrelevant topics due to the lack of side information. Those two drawbacks could lead to human-unfriendly topics that fail to make sense to end users in practice. This issue is further exacerbated in hierarchical cases where a large number of topics and their relevance need to be modeled [27, 12].

In many cases, users with prior domain knowledge are concerned with a specific topic structure, as shown in Fig. 1. Such a topic hierarchy provides semantic common sense among topics and words, which can guide high-quality topic discovery. Therefore, several researches have attempted to exploit

---

[*]Corresponding author

36th Conference on Neural Information Processing Systems (NeurIPS 2022).

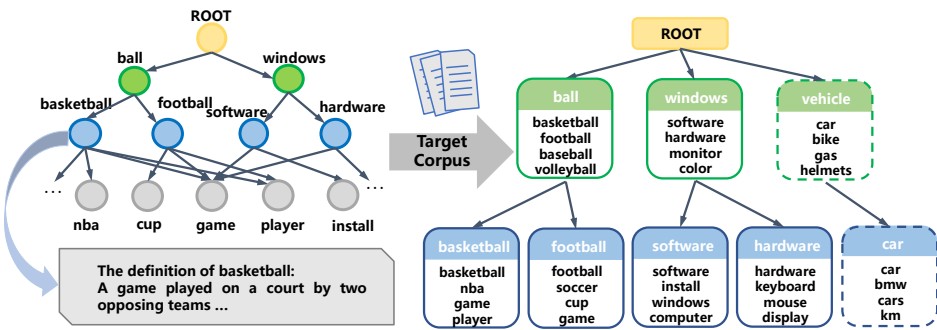

Figure 1: Motivation of the proposed model. We use the pre-specified topic hierarchy as the knowledge graph (left), where topics (colored nodes) and words (gray nodes) are organized into a tree-like taxonomy, and each topic is associated with a brief definition. Given the topic tree and the target corpus, our proposed model aims to *1)* retrieve a set of coherent words for each topic node (right); *2)* explore new topic hierarchies that are missed in the provided topic tree, e.g., the dashed boxes (right).

various prior knowledge into topic modeling to improve topic representation. For example, word correlation-based topic models [1, 30] use must-links and cannot-links to improve the interpretability of learned topics. Word semantic methods [18, 11] allow users to predefine a set of seed words that could appear together in a certain topic to improve topic modeling. Furthermore, knowledge graph-based topic models [17, 34] combine LDA with entity vectors to inject multi-relational knowledge, achieving better semantic coherence. Although the above mentioned methods improve topic coherence in different ways, they only work with shallow topic models and ignore the relationships between topics.

More recently, embedded topic models (ETMs) and their hierarchical variants [10, 38, 32, 12] employ distributed embedding vectors to encode semantic relationship, which have gained growing research interest due to their effectiveness and better flexibility. Viewing words and topics as embeddings in a shared latent space makes it possible for ETMs to integrate prior knowledge in the forms of pretrained word embeddings or semantic regularities [10, 27, 13]. The main idea behind those models is to constrain the topic embeddings under the predefined topic structure. Those models however are built on a strong assumption that the given topic hierarchy is well defined and matched to the target corpus, which is often unavailable in practice. Such mismatched structure might hinder the learning process and lead to sub-optimal results.

To address the above shortcomings, in this paper, we first propose TopicKG, a knowledge graph-guided deep ETM that views topics and words as nodes in the predefined topic tree. As a Bayesian probabilistic model, TopicKG aims to model the document Bag-of-Word (BoW) vector and the given topic tree jointly by sharing the embedding vectors. Specifically, we first adopt a deep document generation model to discover multi-layer document representations and topics. To incorporate the human knowledge in the topic tree and guide the learning of topic hierarchy, a graph generative model is employed, which refines the node embeddings by exploring the belonging relations hidden in topic tree, resulting in more interpretable topics. Besides, to address the mismatch between the given topic tree and target corpus, we further extend TopicKG to TopicKGA, which adopts the graph **A**daptive technique to explore the missing links in topic tree from the target corpora, achieving better document representation. The final revised topic tree is obtained by combining the prior structure of the given topic tree with the added structures learned from the current corpus. Thus the topic tree in our TopicKGA has the ability to reinforce itself constantly according to the given corpus. Finally, both the proposed models are based on variational autoencoders (VAEs) [21] and can be trained by maximizing the evidence lower bound (ELBO), enjoying promising flexibility and scalability.

The main contributions of this paper can be summarized as follows: *1)*, A novel knowledge-aware deep ETM named TopicKG is proposed to incorporate prior domain knowledge into hierarchical topic modeling by accounting for the document and the given topic tree in a Bayesian generative framework. *2)*, To overcome the drawback of the mismatch issue between the prior structure of the provided topic tree and the current corpus, TopicKG is further extended to TopicKGA, which allows to revise the prior structure to better represent the current corpus. *3)*, Besides achieving promising performance on hierarchical topic discovery and document classification task, the proposed models give good flexibility and efficiency, providing a strong baseline for knowledge-based deep TMs.

## 2    Related Work

**Embedded Topic Models:** The recent developed ETM is a new topic-modeling framework in embedding space. Dieng et al. [10] first proposed ETM with the goal of marrying traditional topic models with word embeddings. In particular, ETM regards words and topics as continue embedding vectors and the topic's distribution over words is calculated using the inner product of the topic's embedding and each word's embedding, which provides a natural way to incorporate word meanings into TMs. In constrast to LDA, ETM employs the logistic-normal distribution to estimate the posterior of per-document topic proportion, making it is easier to reparameterization in the inference step. All parameters in ETM are optimized by maximizing its evidence lower bound (ELBO). To explore hierarchical document topical representations, ETM is extended to SawETM [12] with the gamma belief network [42]. SawETM designs the sawtooth connection (SC) to capture the relations between topics at two adjacent layers in the embedding space, resulting in more efficient algorithm for hierarchical topic mining. Note that our proposed TopicKG is built upon SawETM, but different from SawETM that only considers the BoW representation, which may fail to discover high quality topic hierarchies when a large of topics and their relations need to be modeled [27, 13], TopicKG introduces a novel Bayesian generative model where both document and the pre-specific domain knowledge are considered to integrate human knowledge into NTMs, resulting in a new knowledge-aware topic modeling framework.

**Topic Models With Various Knowledge:** Pre-trained language models such as BERT [9] and GPT [6] have been successful in various natural language processing tasks. Pre-trained on huge text, such models can serve as a powerful encoder that outputs contextual semantic embeddings. Behind this idea, combining a TM with BERT via knowledge distillation [16] treats BERT as the teacher model to improve the representation of topic models. CombinedTM [3] concatenates the Bag-of-Word (BoW) vector with the BERT contextual embedding as the final input of a TM, and performs a consistent increase in topic coherence. Despite their improvement, those BERT-based TMs focus on the document latent representation, they ignore the relationship between words and topics. Moreover they require a large amount of memory to load the pre-trained BERT. On the other hand, incorporating knowledge graph (*e.g.*, WordNet [28]) to improve existing NTMs becomes an interesting direction recently. For example, [27] use the category tree (usually with simple and shallow structure) as the supervised information and preserve such relative hierarchical structure in the spherical embedding space. to explore user interested topic structure. [13] view words and topics as the Gaussian distributions and employ the asymmetrical Kullback–Leibler (KL) divergence to measure the directional similarity in the given topic hierarchy. However, computing such KL divergence for each pair is time-consuming that limits its application to large-scale knowledge. Moreover, those knowledge-based models ignore the mismatch problem between the given topic tree and the current corpus, leading to sub-optimal results in practice.

## 3    The Proposed Model

To incorporate human knowledge into deep TMs, we first propose TopicKG that generates the target corpus and the given topic tree in the Bayesian manner. We further extend TopicKG to its adaptive version TopicKGA that allows the topic structure to be refined according to the current corpus, resulting in a better document representation. Below we introduce the details of our proposed model.

### 3.1    Problem Formulation

Consider a corpus containing $J$ documents with a vocabulary of $V$ unique tokens. Each document is represented by a BoW vector $\boldsymbol{x} \in R_+^V$, where $x_v$ denotes the number of occurrences of the $v^{th}$ word. Unlike other TMs that are purely data-driven, TopicKG intends to inject the knowledge graph to improve topic quality. Specifically, we introduce a topic tree $\mathcal{T}$ as our prior knowledge, where each node $e_i \in \mathcal{T}$ denotes a word or a topic. For each topic node, there is a corresponding definition as shown in Fig. 1 . Suppose there are $L + 1$ layers in the topic tree with the bottom word layer and $L$ topic layers, and there are $K_l$ nodes $\mathcal{T}_l : \{e_1^{(l)}, ..., e_{K_l}^{(l)}\}$ at the $l$-th layer, where $K_0 = V$. Generally, for the $k$-th topic node at the $l$-th layer $e_k^{(l)}$, we use $\mathcal{S}(e_k^{(l)})$ to denote the set of its child nodes, and use $\mathcal{C}(e_k^{(l)})$ to denote its key words extracted from the definition sentence. Mathematically, we adopt the binary matrix $\boldsymbol{S}^{(l)} \in \{0, 1\}^{K_{l-1} \times K_l}$ and $\boldsymbol{C}^{(l)} \in \{0, 1\}^{V \times K_l}$, $l = 1, ..., L$, to denote

the belonging relations between two adjacent layers and the links between topics and their concept words, respectively. Guided by this pre-specified topic tree, TopicKG expects to retrieve hierarchical topics from the target corpus that provide a clear taxonomy for the end users.

## 3.2 Knowledge-Aware Bayesian Deep Topic Model

Given the corpus $\mathcal{D}$ and topic tree $\mathcal{T}$, we aim to discover hierarchical document representations and topic hierarchy by jointly modeling the BoW vector $\boldsymbol{x}$, $\boldsymbol{S}^{(l)}$ and $\boldsymbol{C}^{(l)}$, $l = 1, ... L$ in a Bayesian framework. Specifically, we employ the gamma belief network (GBN) of [40] to model the count-value vector $\boldsymbol{x}$, and use the semantic similarity between two node embeddings to generate the topic tree $\mathcal{T}$. The whole generative model with $L$ layers can be expressed as:

$$
\boldsymbol{\theta}_j^{(L)} \sim \text{Gam}(\boldsymbol{\gamma}, 1/c_j^{(L+1)}), \quad \left\{ \boldsymbol{\theta}_j^{(l)} \sim \text{Gam}(\boldsymbol{\Phi}^{(l+1)}\boldsymbol{\theta}_j^{(l+1)}, 1/c_j^{(l+1)}) \right\}_{l=1}^{L-1},
$$

$$
\boldsymbol{x}_j \sim \text{Pois}(\boldsymbol{\Phi}^{(1)}\boldsymbol{\theta}_j^{(1)}) \quad \left\{ \boldsymbol{\Phi}_k^{(l)} = \text{Softmax}(\boldsymbol{\Psi}_k^{(l)}) \right\}_{l=1}^{L}, \quad \left\{ \boldsymbol{\Psi}_{k_1 k_2}^{(l)} = \boldsymbol{e}_{k_1}^{(l-1)T}\boldsymbol{e}_{k_2}^{(l)} \right\}_{k_1=1,k_2=1,l=1}^{K_{l-1},K_l,L},
$$

$$
\left\{ S_{k_1,k_2}^{(l)} \sim \text{Bern}(\sigma(\boldsymbol{e}_{k_1}^{(l-1)T}\boldsymbol{W}\boldsymbol{e}_{k_2}^{(l)})) \right\}_{k_1=1,k_2=1,l=1}^{K_{l-1},K_l,L}, \left\{ C_{vk}^{(l)} \sim \text{Bern}(\sigma(\boldsymbol{e}_v^{(0)T}\boldsymbol{e}_k^{(l)})) \right\}_{v=1,k=1,l=1}^{V,K_l,L},
$$

$$(1)$$

where $\boldsymbol{x}_j$ is generated as in Poisson factor analysis [41], in which $\boldsymbol{x}_j$ is factorized as the product of the factor loading matrix (topics) $\boldsymbol{\Phi}^{(1)} \in \mathbb{R}_+^{K_0 \times K_1}$ and the gamma distributed factor scores (topic proportions) $\boldsymbol{\theta}_j^{(1)} \in \mathbb{R}_+^{K_1}$ under the Poisson likelihood; Then the GBN [42] is applied to explore multi-layer document representations by stacking hierarchical prior in the topic proportions $\boldsymbol{\theta}_j$. The topics at each layer $\boldsymbol{\Phi}^{(l)}$, $l = 1, ..., L$ is calculated by the semantic similarity (*e.g.*, the inner product) between corresponding topic embeddings, followed by a Softmax layer to satisfy the probability simplex: $\sum_{k_{l-1}=1}^{K_{l-1}} \Phi_{k_{l-1}k}^{(l)} = 1$. $\boldsymbol{e}_k^{(l)} \in R^d$ is the embedding vector of $k$-th topic at $l$-th layer ($k$-th node at $l$-th layer $e_k^l$ in $\mathcal{T}$), $d$ is the embedding dimension. $\boldsymbol{e}_v^{(0)}$ is the embedding vector for $v$-th word in vocabulary. $\sigma(\cdot)$ is the sigmoid function and Bern$(\cdot)$ denotes the Bernoulli distribution that is employed to model the two edge types $\boldsymbol{S}^{(l)}$ and $\boldsymbol{C}^{(l)}$. As suggested in [27] and [13], we use asymmetry to capture the directed relations in $\boldsymbol{S}^{(l)}$, while use symmetry to capture the semantic similarity between topics and their concept words. $\boldsymbol{W}$ is a learnable parameter to guarantee the directed structure. Under the Bernoulli likelihood, our TopicKG has the ability to incorporate the topic hierarchical relations in $\mathcal{T}$ into topic modeling by sharing word and topic embeddings with $\boldsymbol{\Phi}$. In particular, for a node pair in $\mathcal{T}$, the embedding semantics of the source and destination nodes need to be similar enough to determine whether there is an edge, which will guide the learning of $\boldsymbol{\Phi}$, resulting in more coherent topics and some appealing model properties as described below.

**The Flexibility of Human Knowledge Incorporation:** As mentioned above, guided by the pre-defined topic tree, TopicKG models $\boldsymbol{x}$, $\boldsymbol{S}^{(l)}$ and $\boldsymbol{C}^{(l)}$, $l = 1, ... L$ jointly, making it possible for the learned hierarchical topics that not only satisfy the interpretable taxonomy structure but also prevent incoherent issues with the help of side information. Besides, the incorporation of topic tree $\mathcal{T}$ is simple and flexible. Firstly, as shown in Eq. 1 and Sec. 4, the plug-and-play module of $\boldsymbol{S}^{(l)}$ and $\boldsymbol{C}^{(lk)}$ can be applied to most of ETMs without changing their model structures, which provides a convenient alternative for introducing side information to TMs. Secondly, the pre-defined $\mathcal{T}$ can be constructed in various ways to encode our beliefs about the graph structure, *e.g.*, the knowledge graph, taxonomy, and hierarchical label tree [34, 17, 27].

**The Relationship Between $\{\boldsymbol{\Phi}^{(l)}\}_{l=1}^{L}$ and $\{\boldsymbol{S}^{(l)}\}_{l=1}^{L}$:** The topic distribution matrix $\boldsymbol{\Phi}^{(l)}$ and the symmetry matrix $\boldsymbol{S}^{(l)}$ are related but play different roles. The sparse structure matrix $\boldsymbol{S}^{(l)}$ is constructed from the topic tree which will be used to guide the learning of the hierarchy of topics, while the learned topics $\boldsymbol{\Phi}^{(l)}$ contains more specific knowledge extracted from the text corpus, and can be viewed as the dense version of $\boldsymbol{S}^{(l)}$. In other words, the topics learned from TopicKG retains the semantic structure of $\boldsymbol{S}^{(l)}$ and enriches itself by the current corpora.

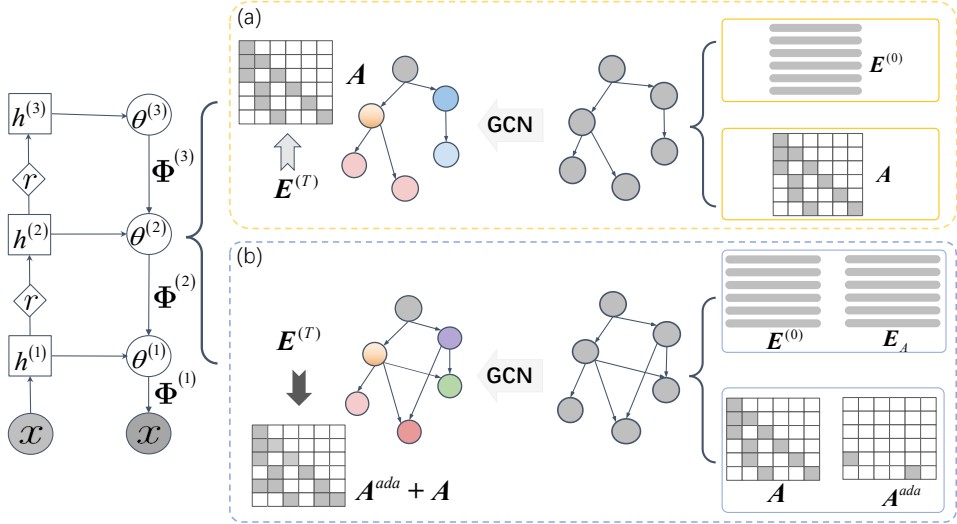

Figure 2: Overview of the proposed framework. It jointly models the topic tree and document via the shared node embeddings $\boldsymbol{E}^{(T)}$. The proposed two models TopicKG (a) and TopicKGA (b) share the same document generative module (left part) while with different graph modelings.

## 3.3 Inference Network of TopicKG

The inference network of the proposed model is built around two main components: Weibull upward-downward variational encoder and the GCN-based topic aggregation module. The former aims to infer the topic proportions given the document, and the latter updates the node embedding by aggregating the neighbor information via the GCN layer.

**Weibull Upward-Downward Variational Encoder:** To approximate the posterior of the topic proportions $\{\boldsymbol{\theta}_j^{(l)}\}$, like most of VAE-based methods, we define the variational distribution $q(\boldsymbol{\theta}_j|\boldsymbol{x}_j)$, which can be further factorized as [31]:

$$q(\boldsymbol{\theta}_j|\boldsymbol{x}_j) = q(\boldsymbol{\theta}_j^{(L)}|\boldsymbol{x}_j) \prod_{l=1}^{L-1} q(\boldsymbol{\theta}_j^{(l)}|\boldsymbol{\theta}_j^{(l+1)}, \boldsymbol{x}_j).$$

In practice, it first obtains the latent feature by feeding the input $\boldsymbol{x}_j$ into a residual upward neural networks: $\boldsymbol{h}_j^{(l)} = \boldsymbol{h}_j^{(l-1)} + f_{\boldsymbol{W}_h^{(l)}}^{(l)}(\boldsymbol{h}_j^{(l-1)})$, where $l = 1, ..., L$, $\boldsymbol{h}_j^{(0)} = \boldsymbol{x}_j$, $f_{\boldsymbol{W}_h^{(l)}}^{(l)}(\cdot)$ is a two layer fully connected network parameterized by $\boldsymbol{W}_h^{(l)}$. To complete the variational distribution, we adopt the Weibull downward stochastic path [37]:

$$q(\boldsymbol{\theta}_j^{(l)}|\boldsymbol{\Phi}^{(l+1)}, \boldsymbol{h}_j^{(l)}, \boldsymbol{\theta}_j^{(l+1)}) = \text{Weibull}(\boldsymbol{k}_j^{(l)}, \boldsymbol{\lambda}_j^{(l)})$$

$$\boldsymbol{k}_j^{(l)} = \text{Softplus}(f_k^{(l)}(\boldsymbol{\Phi}^{(l+1)}\boldsymbol{\theta}_j^{(l+1)} \oplus \hat{\boldsymbol{k}}_j^{(l)})), \quad \boldsymbol{\lambda}_j^{(l)} = \text{Softplus}(f_\lambda^{(l)}(\boldsymbol{\Phi}^{(l+1)}\boldsymbol{\theta}_j^{(l+1)} \oplus \hat{\boldsymbol{\lambda}}_j^{(l)})) \quad (2)$$

$$\hat{\boldsymbol{k}}_j^{(l)} = \text{Relu}(\boldsymbol{W}_k^{(l)}\boldsymbol{h}_j + \boldsymbol{b}_k^{(l)}), \quad \hat{\boldsymbol{\lambda}}_j^{(l)} = \text{Relu}(\boldsymbol{W}_k^{(l)}\boldsymbol{h}_j + \boldsymbol{b}_k^{(l)}),$$

where $\oplus$ denotes the concatenation at topic dimension, and we use the Softplus to make sure the positive Weibull shape and scale parameters. $f$ is a single layer fully connected network, correspondingly. The Weibull distribution is chosen mainly because it is reparameterizable and the KL divergence from the gamma to Weibull distributions has an analytic expression [37, 15].

**GCN-Based Topic Aggregation:** Attracted by the excellent ability of graph neural network (GCN) [22, 23] in propagating graph structure information, and the predefined topic tree can be viewed as a directed graph. We thus construct a deterministic aggregating module for node embeddings with GCN:

$$\boldsymbol{E}^{(t)} = \text{GCN}(\tilde{\boldsymbol{A}}, \boldsymbol{E}^{(t-1)}), t = 1, ..., T \quad (3)$$

where $T$ is the number of GCN layer, $\boldsymbol{E}^{(0)} \in R^{d \times N}$ is the node embedding matrix, and $N = \sum_{l=0}^{L} K_l$. $\tilde{\boldsymbol{A}} = \boldsymbol{D}^{-\frac{1}{2}} \boldsymbol{A} \boldsymbol{D}^{-\frac{1}{2}}$ is the normalized adjacent matrix with degree matrix $\boldsymbol{D}$, $e.g.$, $\boldsymbol{D}_{ii} = \sum_{j=1}^{N} \boldsymbol{A}_{ij}$, $\boldsymbol{A} \in \{0,1\}^{N \times N}$ is the adjacent matrix of the topic tree $\mathcal{T}$. Unlike previous ETMs that view embeddings as independent learnable parameters, the GCN module in TopicKG updates the node embeddings by considering their child-level topics and related words, resulting in more meaningful topic embeddings. We calculate $\{\boldsymbol{\Phi}^{(l)}\}_{l=1}^{L}$ via the updated node embeddings:

$$\boldsymbol{\Phi}_k^{(l)} = \text{Softmax}(\boldsymbol{e}^{(T)^{(l-1)^T}} \boldsymbol{e}_k^{(T)^{(l)}}) \tag{4}$$

### 3.4 TopicKG With Adaptive Structure

One of the main assumptions in the previous section is that the pre-defined topic tree is helpful for the current corpus and the corresponding edges are highly reliable. However, this is generally unrealistic in practical applications, as $\mathcal{T}$ may be (i) noisy, (ii) built on an ad hoc basis, (iii) not closely related to the topic discovering task [14, 8]. Consequently, we further propose TopicKGA that overcomes the above mismatching issue and revises the structure of the given topic tree based on the corpus at hand.

In detail, TopicKGA first randomly initializes a learnable node embedding dictionaries $\boldsymbol{E}_A \in R^{d \times N}$ for all nodes in $\mathcal{T}$. Then an adaptive graph $\boldsymbol{A}^{\text{ada}}$ is generated based on $\boldsymbol{E}_A$ using the certain kernel function $k : R^d \times R^d \longrightarrow R$:

$$\tilde{\boldsymbol{A}}^{\text{ada}} = \text{Softmax}(k(\boldsymbol{E}_A, \boldsymbol{E}_A)) \tag{5}$$

Here, we choose the consine similarity to define our kernel function: $k(\boldsymbol{E}_A, \boldsymbol{E}_A) = \frac{\boldsymbol{E}_A \boldsymbol{E}_A^T}{||\boldsymbol{E}_A|| ||\boldsymbol{E}_A||}$, and Softmax function is used to normalize the adaptive matrix. Note that, instead of generating $\boldsymbol{A}^{\text{ada}}$, we directly generate its normalized version to avoid unnecessary calculations [2]. During training, $\boldsymbol{E}_A$ will be updated automatically to learn the hidden dependencies which are ignored by $\boldsymbol{A}$. Following [36], the final revised adjacency matrix is formed as (here we still use $\tilde{\boldsymbol{A}}$ to denote the revised graph for convenience):

$$\tilde{\boldsymbol{A}} = \tilde{\boldsymbol{A}} + \tilde{\boldsymbol{A}}^{\text{ada}} \tag{6}$$

which will enhance the GCN in Eq. 3 by replacing the adjacency matrix with the revised one.

## 4 Training

As a deep ETM, TopicKG intends to learn the latent document-topic distribution and the deterministic hierarchical topic embeddings. Like other ETMs, the posterior of the topic proportions and topic embeddings are intractable to compute. We thus derive an efficient algorithm for approximating the posterior with amortized variational inference [5], which makes the proposed model flexible for downstream task. The resulting algorithm can either use pre-trained word embeddings, or train them from scratch. The ELBO of the proposed model can be expressed as:

$$\begin{aligned}
\mathcal{L} = \sum_{j=1}^{J} \text{E}_{q(\boldsymbol{\theta}|\boldsymbol{x}_j)}[\log p(\boldsymbol{x}_j|\boldsymbol{\Phi}^{(1)}, \boldsymbol{\theta}_j^{(1)})] + \beta \sum_{l=1}^{L} \sum_{k_2=1}^{K_l} (\sum_{k_1=1}^{K_{l-1}} \log p(S_{k_1 k_2}^{(l)})|\boldsymbol{e}_{k_1}^{(l-1)}, \boldsymbol{e}_{k_2}^{(l)}) \\
+ \sum_{v=1}^{V} \log p(C_{v k_2}^{(l)}|\boldsymbol{e}_v^{(0)}, \boldsymbol{e}_{k_2}^{(l)})) - \sum_{j=1}^{J} \sum_{l=1}^{L} \text{KL}(q(\boldsymbol{\theta}_j^{(l)})||p(\boldsymbol{\theta}_j^{(l)}|\boldsymbol{\Phi}^{(l+1)}, \boldsymbol{\theta}_j^{(l+1)})))
\end{aligned} \tag{7}$$

It consists of three main parts: the expected log-likelihood of $\boldsymbol{x}$ (first term), the concept structure log-likelihood (the two middle terms), and the KL divergence from prior $p(\boldsymbol{\theta}_j^{(l)})$ to $q(\boldsymbol{\theta}_j^{(l)})$. The graph weight $\beta$ denotes the belief about the predefined topic tree. Notably, the two middle terms distinguish the proposed models from previous deep ETMs. On the one hand, it acts as a regularization that constrains the embedding vectors to conform to the provided prior structure, and on the other hand, it provides an alternative for ETM to introduce side information to improve its interpretability.

**Annealed Training for TopicKGA:** In TopicKGA, the structure of the revised graph changes during the training. To incorporate the new edges that are inferred from the target corpus, we develop an

annealed training algorithm for TopicKGA. Specifically, we update $\boldsymbol{S}^{(l)}$ and $\boldsymbol{C}^{(l)}$ in Eq. 7 every $M$ iterations:

$$S_{k_1 k_2}^{(l)} = \begin{cases} 1, & \text{if} \quad \tilde{A}_{k_1 k_2}^{(l)} > s; \\ 0, & else \end{cases} \quad C_{v k_2}^{(l)} = \begin{cases} 1, & \text{if} \tilde{A}_{v k_2}^{(l)} > s; \\ 0, & else \end{cases} \quad (8)$$

where $s$ is the threshold, and $\tilde{\boldsymbol{A}}^{(l)}$ is the corresponding sub-block in $\tilde{\boldsymbol{A}}$. As $\boldsymbol{E}_A$ becomes stable, the structure of the graph will converge to a good blueprint that balances the prior graph and the current corpus. We summarize all training algorithm in Appendix.

## 5 Experiment

In this section, we conduct extensive experiments on several benchmark text datasets to evaluate the performance of the proposed models against other knowledge-based TMs, in terms of topic interpretability and document representations. The code is available at https://github.com/wds2014/TopicKG.

### 5.1 Corpora

Our experiments are conducted on four widely used benchmark text datasets, varying in scale. The datasets include 20 Newsgroups (20NG) [24], Reuters extracted from the Reuters-21578 dataset, R8, and Reuters Corpus Volume 2 (RCV2) [25]. R8 is a subset of Reuters that collected from 8 different classes. For the multi-label RCV2 dataset, we follow previous works [29] in which documents with a single label at the second level topics are left, resulting in 0.1M documents totally. Both R8 and RCV2 are already pre-processed. For other two datasets, we tokenize and clean text by excluding standard stop words and low-frequency words appearing less than 20 times [35]. The statistics of the preprocessed datasets are summarized in Appendix.

### 5.2 Baselines

To demonstrate the effectiveness of incorporating human knowledge into deep TMs, we consider several baselines for a fair comparison, including representative ETMs and recent knowledge-based topic models, described as follows: *1)*, **ETM** [10], the first neural embedded topic model that assumes topics and words live in the same embedding space. The topic distributions in ETM are calculated by the inner product of the word embedding matrix and the topic embedding. *2)*, **CombinedTM** [3], a BERT-based neural topic model that uses the pre-trained BERT as its contextual encoder. We use it as our BERT baseline. *3)*, **SawETM** [12], a hierarchical ETM that employs the Poisson and gamma distributions to model the BoW vector and the latent representation, respectively. *4)*, **JoSH** [27], a knowledge-based hierarchical topic mining method that uses category hierarchy as the side information and employs the EM algorithm to learn the spherical tree and text embedding. *5)*, **TopicNet** [13], a semantic graph guided topic model that views the words and topics as the Gaussian distributions and uses the KL divergence to regularize the structure of the pre-defined tree. For all baselines, we employ their official codes and default settings obtained from their release repositories.

### 5.3 External Knowledge

Since the learning of TopicKG involves a pre-defined topic tree, some generic knowledge graphs such as WordNet [28] provide a convenient way to bring in external knowledge. WordNet is a large lexical database that groups semantically similar words into synonym sets, these sets are further linked by the hyponymy relation. For example, in WordNet the category furniture includes bed, which in turn includes bunkbed. With these relations, the topic structure can be easily defined according to certain heuristic rules. Specifically, for each dataset we first get the word intersection of the vocabulary and WordNet, the chosen words are then considered as leaf nodes of the topic hierarchy. Afterwards, each leaf node can continuously find its ancestor nodes based on hyponymy relations and finally all leaf nodes converge to the same root node, resulting in an adaptive topic structure.

### 5.4 Settings

For all experiments, we set the embedding dimension as $d = 50$, the knowledge confidence hyper-parameter as $\beta = 50.0$, the threshold as $s = 0.4$. Empirically, we find that the above settings work

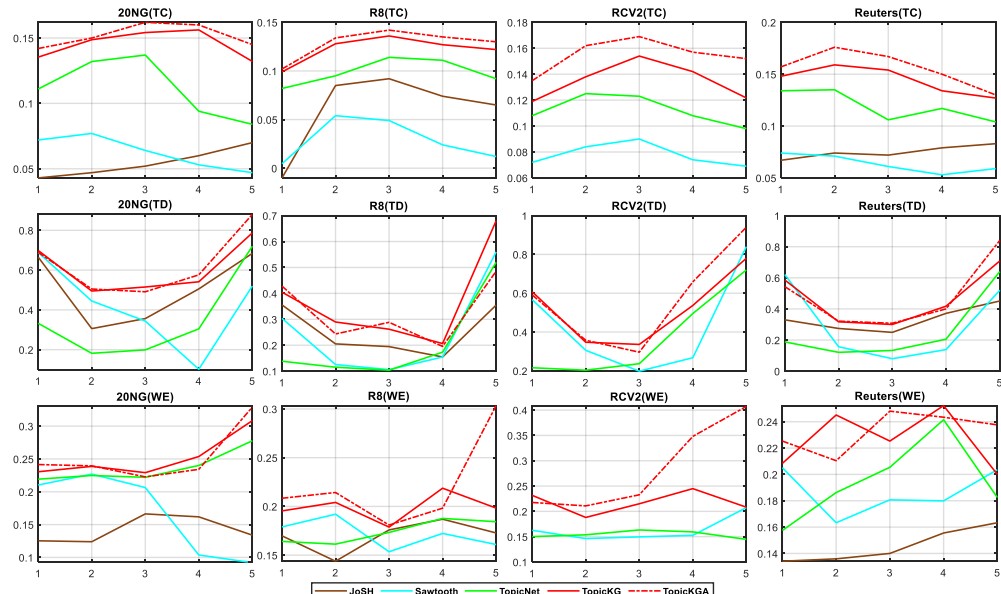

Figure 3: Topic coherence (TC, top row), topic diversity (TD, middle row), and word embedding coherence (WE, bottom row) results for various deep topic models on four datasets. In each subfigure, the horizontal axis indicates the layer index of the topics. For all metrics, higher is better.

well for all datasets, we refer readers to the appendix for more detailed analysis. We initialize the node embedding from the Gaussian distribution $\mathcal{N}(0, 0.02)$. We set the batch size as 200 and use the AdamW [26] optimizer with learning rate 0.01. We generate a 7-layer topic hierarchy for each dataset with the method described in Sec. 5.3. From the top to bottom, the number of topics at each layer are: 20NG: [1,2,11,66,185,277,270]; R8: [1,2,11,77,287,501,547]; RCV2: [1,2,11,68,263,469,562]; Reuters: [1,2,11,78,300,526,584]. For the single-layer methods (ETM and CombinedTM), we set the number of topics as the same of the first layer of deep models. For all methods, we run the algorithms in comparison five times by modifying only the random seeds and calculate the mean and standard deviation. All experiments are performed on an Nvidia RTX 3090-Ti GPU and our proposed models are implemented with PyTorch.

Table 1: Micro F1 and Macro F1 score of different models on three datasets. The digits in brackets indicate the number of layers. Micro F1 /Macro F1.

| Model | 20NG | R8 | RCV2 |
|---|---|---|---|
| ETM | 50.25 ±0.42 / 47.44 ±0.21 | 88.10 ±0.45 / 59.67 ±0.24 | 68.63 ±0.15 / 24.40 ±0.11 |
| CombinedTM | 56.43 ±0.14 / 54.95 ±0.11 | 93.69 ±0.09 /84.14 ±0.10 | 84.85 ±0.11 / 51.47 ±0.21 |
| Sawtooth(3) | 52.41 ±0.08 / 51.53 ±0.10 | 90.04 ±0.15 / 78.84 ±0.21 | 82.54 ±0.11 / 49.25 ±0.10 |
| TopicNet(3) | 55.16 ±0.22 / 54.78 ±0.34 | 89.95 ±0.17 / 64.15 ±0.16 | 84.15 ±0.25 / 50.37 ±0.22 |
| TopicKG(3) | 55.73 ±0.15 / 54.48 ±0.08 | 93.6 ±0.05 / 83.32 ±0.07 | 84.75 ±0.16 / 50.51 ±0.41 |
| TopicKGA(3) | **58.63** ±0.15 / **57.90** ±0.10 | **93.70** ±0.52 / **84.50** ±0.11 | **85.34** ±0.14 / **52.35** ±1.10 |
| ETM | 47.79 ±0.12 / 44.19 ±1.01 | 86.54 ±0.84 / 59.88 ±1.11 | 63.77 ±0.14 / 21.44 ±1.04 |
| CombinedTM | 58.16 ±0.15 / 58.10 ±0.10 | 93.50 ±0.13 /84.84 $pm$0.11 | 82.91 ±0.11 / 48.17 ±0.05 |
| Sawtooth(7) | 53.71 ±0.11 / 53.02 ±0.47 | 92.86 ±0.07 / 82.54 ±0.41 | 82.46 ±0.15 / 49.34 ±0.34 |
| TopicNet(7) | 56.13 ±0.19 / 55.41 ±0.39 | 90.65 ±0.00 / 66.57 ±0.00 | 82.81 ±0.00 / 49.44 ±0.00 |
| TopicKG(7) | 56.32 ±0.12 / 57.35 ±0.04 | 94.04 ±0.12 / 85.04 ±0.11 | 82.48 ±0.11 / 48.24 ±0.09 |
| TopicKGA(7) | **60.04** ±0.34 / **59.12** ±0.13 | **94.10** ±0.08 / **85.50** ±0.10 | **83.08** ±0.23 / **50.50** ±0.08 |

## 5.5 Topic Interpretability

Generally, topic models are evaluated based on perplexity. However, perplexity on the held-out test is not an appropriate measure of the topic quality and sometimes can even be contrary to human judgements [7, 34]. To this end, we instead adopt three common metrics, including Topic coherence (**TC**), Topic diversity (**TD**) and word embedding topic coherence (**WE**), to evaluate the learned topics from various aspects [10, 3]. TC measures the average Normalized Pointwise Mutual Information

(NPMI) over the top 10 words of each topic, and a higher score indicates more interpretable topics. TD denotes the percentage of unique words in the top 25 words of the selected topics. WE, as its name implies, provides an embedded measure of how similar the words in a topic are. WE is calculated by the average pairwise cosine similarity of the word embeddings of the top-10 words in a topic. We report the topic quality results at the first five layer in Fig. 3 [2]. We here focus on topic hierarchy and ignore the single layer methods. We find that: *1)*, Knowledge-based methods including JoSH, CombinedTM, TopicNet and our proposed models are generally better than other likelihood-based ETMs, which illustrates the benefit of incorporating side information; *2)*, Our proposed TopicKG and TopicKGA achieve higher performance than other knowledge-based models in most cases, especially on TC and WE tasks, which means our proposed model prefer to mine more coherent topics while achieving comparable TD. We attribute this to the joint topic tree likelihood that provides an efficient alternative to integrate human knowledge into ETMs. *3)*, Compared to TopicKG, its adaptive version TopicKGA discovers better topics at higher layers. It is not surprise that TopicKGA gives greater robustness and flexibility by refining the topic structure according to the current corpus.

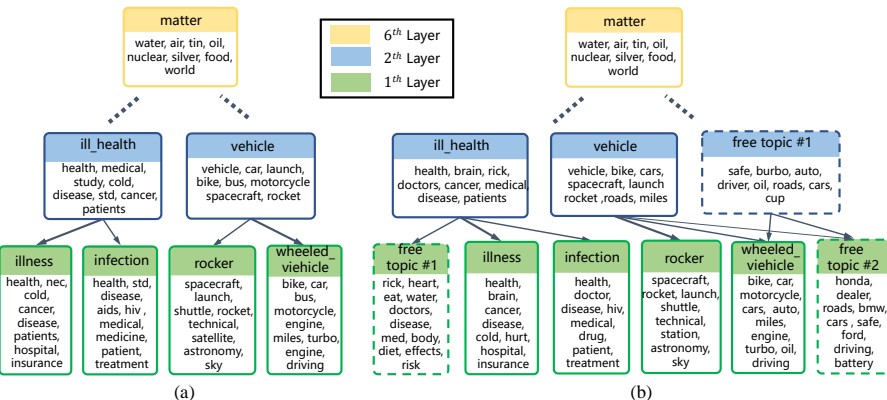

Figure 4: The learned hierarchical topic structure from TopicKG (a) and TopicKGA(b) on 20NG. Each topic box contains its concept name and the corresponding keywords. Topics in dashed box (free topics) are newly added by TopicKGA from the target corpus. The thickness of the arrow represents the relation weight and different colors denote different layers.

## 5.6 Document Classification

Besides the topic quality evaluation, we also use document classification to compare the extrinsic predictive performance. In detail, we collect the inferred topic proportions, $e.g.$, $\theta^{(1)}$, and than apply logistic regression to predict the document label. We report the Micro F1 and Macro F1 score on 20NG, R8 and RCV2 datasets in Table. 1. Overall, the proposed models outperform the baselines on all corpora, which confirms the effectiveness of our innovation of combining human knowledge and ETMs in improving document latent representations. Moreover, with the revised topic structure fine-tuned to the current corpus, TopicKGA surpasses other knowledge graph fixation methods with significant gaps. This conclusion is consistent with one of our motivations that the predefined concept tree may not match the target corpus, leading to suboptimal document representations.

## 5.7 Qualitative Analysis

We visualize the learned topic hierarchies of TopicKG and TopicKGA on 20NG in Fig. 4(a-b), respectively. Each topic box consists of the pre-specified concept name on the top bar and its keywords listed in the bottom content. We can observe that *1)*, The mined keywords are highly relevant to the corresponding topics, providing a clean description of their concepts. *2)*, Guided by human knowledge, the connections between topics at two adjacent layers are highly interpretable, resulting in human-friendly topic taxonomies. *3)*, More interestingly, to further verify the adaptive ability of our proposed TopicKGA, we manually added several free topics (dashed boxes in Fig. 4(b)) to each layer of the given topic tree, which can be viewed as the missing topics in the predefined

---

[2]We can't report the results of JoSH on RCV2 as JoSH requires the sequential text, but only the BoW form is available for RCV2.

knowledge and need to be learned from the target corpus. We find that our TopicKGA indeed has the ability to capture the missing concepts, and revise the prior graph to match the current corpus.

## 5.8 Time Efficiency

To demonstrate the time efficiency of incorporating human knowledge into TMs, we run TopicKG, TopicKGA and TopicNet (as they are implemented in PyTorch, JoSH is in C) on RCV2 and 20NG dataset with a 7-layer topic tree. Fig. 5 shows the Negative log-likelihood of documents $x$, which shows that the proposed models not only have faster learning speed than TopicNet, but also achieve better reconstruction performance on both small and large corpus. This result illustrates the efficiency of the introduced graph likelihood which is important in real-time applications.

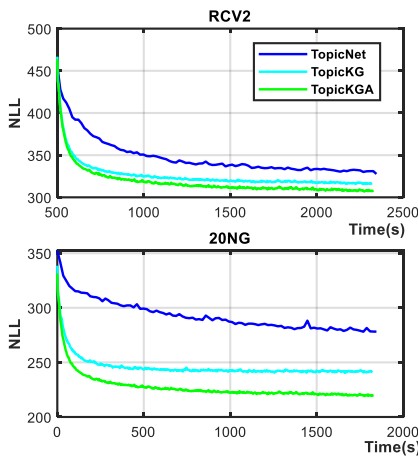

Figure 5: Negative log-likelihood (NLL) curves of various methods on RCV2 and 20NG datasets.

## 6    Conclusion

We develop an efficient Bayesian probabilistic model that integrates pre-specified domain knowledge into hierarchical ETMs. The core idea is the joint document and topic tree modeling with the shared word and topic embeddings. Besides, with the graph adaptive technique, TopicKGA has the ability to revise the given prior structure according to the target corpus, enjoying robustness and flexibility in practice. Extensive experiments show that the proposed models outperform competitive methods in term of both topic quality and document classification task. Moreover, thanks to the efficient knowledge incorporation algorithm, our proposed models achieve faster learning speed than other knowledge-base models.

## Acknowledgements

This work was supported in part by the National Natural Science Foundation of China under Grant U21B2006; in part by Shaanxi Youth Innovation Team Project; in part by the 111 Project under Grant B18039; in part by the Fundamental Research Funds for the Central Universities QTZX22160.

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
