# Appendix of "Knowledge-Aware Bayesian Deep Topic Model"

## A   Detailed discussions of our work

### A.1   Limitations

We in this paper propose a Bayesian generative model for incorporating prior human knowledge into deep topic models, named TopicKG. TopicKG jointly models the provided topic tree $\mathcal{T}$ and the current corpus $\mathcal{D}$ in a Bayesian framework by sharing the word embeddings and topic embeddings. Besides, TopicKG employs a Weibull upward-downward variational encoder to infer hierarchical document representations and a GCN-based topic aggregation module to account for the relations between nodes in the topic tree. To address the mismatch issues between the provided topic tree and the target corpus, TopicKG is further extended into TopicKGA which uses the graph adaptive technique to revise the tree structure according to the current learning state. Despite the promising results of those two proposed models, a main limitation of this work is the form of knowledge. TopicKG and TopicKGA are designed for the prior domain knowledge that contains the relations of entities. While most existing knowledge graphs (e.g. WordNet[Miller, 1995], DBpedia[Auer et al., 2007], Freebase[Bollacker et al., 2008]) meet this requirement, the recent pre-trained language model, such as BERT[Devlin et al.], XLNET[Yang et al., 2019] can be viewed as another form of knowledge. How to combine deep topic models with such pre-trained language model remains a challenge in the topic modelling community. This is beyond the scope of this paper and we leave it for future research.

### A.2   Negative societal impacts

The proposed model improves deep topic models by incorporating side information, It not only achieves state-of-the-art performance in document representation tasks (e.g., document classification), but also discovers coherent and diverse topics, which can be applied in many natural language processing (NLP) tasks, such as document generation, dialogue systems and text mining. Note that technology is neither inherently helpful nor harmful. It is simply a tool and the real effects of technology depend upon how it is wielded. For example some merchants or advertisers may use TopicKG to recommend articles with specific contents of interest to their users, which may induce users to buy some unnecessary items. This negative impact can be avoided by strengthening data regulation to avoid false advertising.

## B   Datasets

To evaluate the performance of the proposed models, we conduct experiment on four widely-used benchmark text datasets, varying in scale.

- **20NG**[1]: The 20 Newsgroups data set is a collection of 18,864 newsgroup documents, partitioned (nearly) evenly across 20 different newsgroups. We follow the default training/testing split and choose the top 2000 words after removing stop words and low-frequency words. The average length of document is 108.75.

---

[1]http://qwone.com/ jason/20Newsgroups

- **R8**: R8 is a subset of Reuters with 8 different classes. There are 5,485 training samples, and 2,189 test samples. The average length of R8 is 65.72.
- **Reuters** [2]: The documents in the Reuters collection appeared on the Reuters newswire in 1987. Here we only use Reuters on topic quality task. the average length of Reuters is 74.14.
- **RCV2**[3]: Reuters Corpus Volume v2 is a benchmark dataset on text categorization. It is a collection of newswire articles produced by Reuters in 1996-1997. The average length of RCV2 is 52.82.

A summary of dataset statistics is shown in Table 1.

| Dataset | $J_{\textbf{all}}$ | $J_{\textbf{train}}$ | $J_{\textbf{test}}$ | $C$ | $V$ | $Avg.N$ |
|---------|--------|----------|---------|-----|-----|---------|
| 20NG   | 18,864  | 11,314   | 7,532   | 20  | 2,000 | 108.75 |
| Reuters | 11,367 | 11,367   | /       | /   | 8,817 | 74.14  |
| R8     | 7,674   | 5,485    | 2,189   | 8   | 7,688 | 65.72  |
| RCV2   | 150,737 | 100,899  | 49,838  | 52  | 7,282 | 82.82  |

Table 1: Summary statistics of the datasets, where $J$ denotes the number of documents, $C$ the number of classes, $V$ the vocabulary size and $Avg.N$ the average length of documents in the corpus, respectively.

## C   Training algorithm

We summary the training algorithm as bellow:

---
**Algorithm 1** Training algorithm for our proposed models
---
**Input**: training documents $\mathcal{D}$, topic tree $\mathcal{T}$, topic number of each layer $K$, hyperparameter $\beta$, $s$ and $M$.
**Initialize**: node embedding $\boldsymbol{E}$ and $\boldsymbol{E}_A$, all learnable parameters.
**for** iter = 1,2, $\cdots$ **do**
    Sample a batch of $J$ input documents; and feed them into the variational encoder to infer the latent representation $\boldsymbol{\theta}$ with Eq.2 in the manuscript;
    **if** TopicKGA **then**
        Calculate the normalized $\tilde{A}$ with Eq.5 in the manuscript;
    **end if**
    With the GCN-based topic aggregation module, update all node embeddings with Eq.3 and calculate the global parameters $\boldsymbol{\Phi}$;
    **if** TopicKGA **and** everay M iterations **then**
        Update $\boldsymbol{S}^{(l)}$ and $\boldsymbol{C}^{(l)}$ with Eq.8;
    **end if**
    Compute the loss with Eq.7, and update all learnable parameters.
**end for**
---

## D   Hypeparameter sensitivity

We fix the knowledge confidence weight $\beta = 50.0$ and the threshold $s = 0.4$ in the previous experiments. Bellow we analyze the performance of the proposed models at different hyperparameter setting and report the Micro F1 score and WE on 20NG, R8, and RCV2 datasets in Fig. 1 and Fig. D, respectively. We find that 1), Overall, by accounting for both concept structure and document likelihood, HFTM achieves better document representation and topic discovery than only considering the document ($e.g.$, $\beta = 0$). 2), TopicKGA is insensitive to graph generation shreshold $s$ and has a wide tolerance. 3), One can obtain better results by finetuning $\beta$ and $s$ for each dataset.

---
[2]https://kdd.ics.uci.edu/databases/reuters21578/reuters21578.html
[3]https://trec.nist.gov/data/reuters/reuters.html

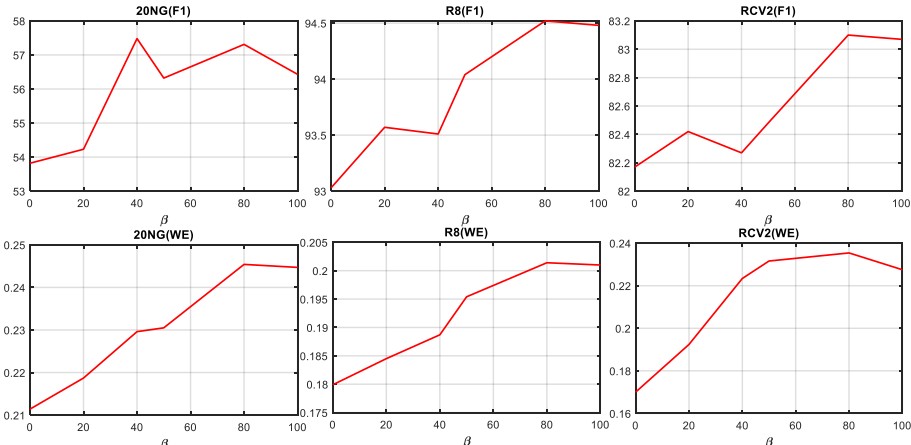

Figure 1: Micro F1 score (F1, top row) and word embedding coherence (WE, bottom row) on 20NG, R8 and RCV2 datasets with different $\beta$.

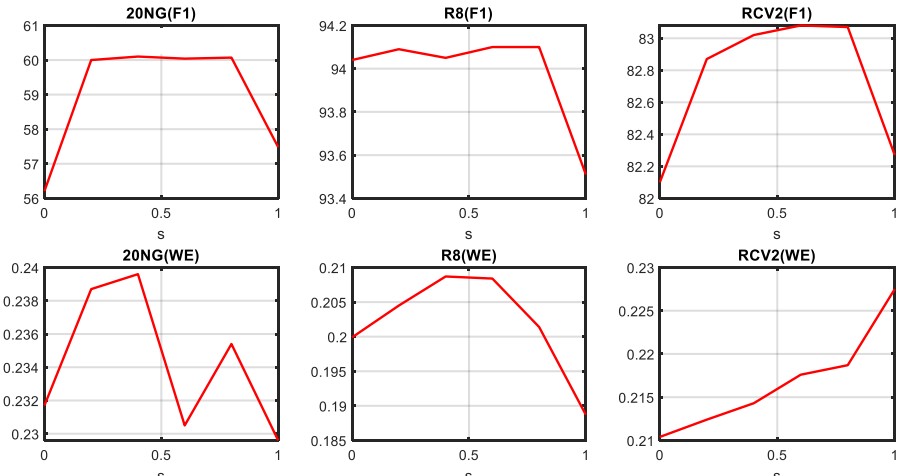

Figure 2: Micro F1 score (F1, top row) and word embedding coherence (WE, bottom row) on 20NG, R8 and RCV2 datasets with different $s$.