# OpenReview forum: "Knowledge-Aware Bayesian Deep Topic Model"
_NeurIPS.cc/2022/Conference — NeurIPS 2022 Accept_

### Official Review · Reviewer_3oGT · 2022-07-10

**Rating:** 7
**Confidence:** 3
**Soundness:** 4 excellent
**Presentation:** 3 good
**Contribution:** 3 good

**Summary:**

The paper proposes a topic modeling approach that extends previous work on Gamma Belief Network [1] to jointly model a knowledge graph provided as input to guide the topics. The graph structure and the keywords representing each graph node is integrated into the generative process. The graph node embedding is modelled using a graph neural networks, which is jointly modelling during the inference process. This allows data driven changes to the knowledge graph, as illustrated as the addition of new topics in the experimental results.

[1] Sawtooth Factorial Topic Embeddings Guided Gamma Belief Network. Duan et al. ICML 2021


**Questions:**

Other than addition of new nodes, does the graph structure evolve? e.g., are edges added or removed?

**Limitations:**

- as rightly pointed out in the appendix, it will be interesting to investigate how the knowledge in large language models can be used in such topic modeling works

**Strengths And Weaknesses:**

Strength
- the proposed approach of guiding a topic model with a knowledge graph seems more natural than previous work such as [1], and in their evaluation, was shown to achieve better evaluation results compared to previous work.

- the proposed model allowed the graph structure to evolve according to the data in a variant called TopicKGA, which is the best performing model

- they showed that their approach converges better than TopicNet [1]


Weaknesses
- more ablation could be done on how GNN contributes in the model. The dynamic structure of the topic graph is only shown in the addition of new topics. Was there any changes in the graph structure other than addition of nodes?

- using wordnet as the knowledge graph seems too general a way to guide a topic model. Knowledge graphs such as those derived from Wikidata or KG more specifically customized to user needs could lead to more practical usage.


[1] TopicNet: Semantic Graph-Guided Topic Discovery. Duan et al. 2021.

---

> ### Author Response · Authors · 2022-08-02
> **Response to Reviewer 3oGT**
>
> Thank you for your time and the positive feedback.
> We have updated the rebuttal revision, with the major changes marked in blue.
> Below we respond to your questions/criticisms.
>
> **w1**
>
> We follow your advice and replace the GCN layer with the MLP layer as the ablation study. We report the topic quality and document classification result below:
>
> | Method        | TD     | TC     | WE     | Micro F1 |
> |:-------------:|:------:|:------:|:------:|:--------:|
> | TopicKG\-MLP  | 0.697 | 0.133 | 0.221 | 55.55   |
> | TopicKG       | 0.698 | 0.135 | 0.230 | 55.73    |
> | TopicKGA\-MLP |  0.688      |   0.141     | 0.237   |    57.92      |
> | TopicKGA      |    0.689    |     0.142   | 0.241    |     58.63     |
>
> From which we can see that the GCN layer in our proposed model gives a consistent improvement on both topic quality and document representation. This demonstrates the effectiveness of the proposed GCN-based topic aggregation module. We will report results on all datasets in the final revision.
>
> To address the mismatch issue between the prior topic tree and the target corpora, the proposed TopicKGA allows the topic structure to be fine-tuned according to the target corpora. Mathematically, it allows adding new edges and nodes to the topic graph. Fig.4 in the manuscript empirically shows both two examples (not only the case of the new topics).
>
> **w2**
>
> WordNet is a common-used knowledge graph in topic models and it contains many semantic relations between words, such as the synonym and hyponymy.  We note that our proposed model is knowledge-agnostic, and users are allowed to extract topic tree from other knowledge bases or provide their topic tree.
>
> **Q1**
>
> In fact, TopicKGA employs the annealed training algorithm to add new knowledge to the prior topic tree (Eq.8 in the manuscript). The core assumption behind TopicKGA is that the relations in the provided topic tree are really existing (true), it will not remove any nodes and edges during the training. It will complete the prior topic structure according to the current corpora by adding the new nodes and edges, which can be viewed as the new knowledge inferred from the target documents.

---

### Official Review · Reviewer_1Lsj · 2022-07-11

**Rating:** 5
**Confidence:** 2
**Soundness:** 2 fair
**Presentation:** 2 fair
**Contribution:** 2 fair

**Summary:**

The goal of this paper is to learn topic hierarchies by deriving knowledge from a pre-existing hierarchy of topics. This pre-existing hierarchy does not have topical words in the nodes.

To this end, the authors develop a Bayesian deep topic model. In the topic model, the Bayesian component is the topic tree that is used as a piece of prior knowledge of the model. The deep part is derived from the Gamma Belief Network framework.  Later, the authors used GCN-based methods to model the hierarchies. Parameter estimation is then presented using amortized variational inference model. Experimental results demonstrate that the model improves upon existing methods.

**Questions:**

Can the authors mention some real-world applications where this work could be useful? How and why this model can be applied to real-world tasks? Section 5.8 only uses the 20NG dataset that is very small.

**Ethics Review Area:**

["I don’t know"]

**Limitations:**

The key issue is that the model relies on an external knowledge base. One might need knowledge bases in different domains to use this model.

**Strengths And Weaknesses:**

The key strength lies in modelling topic hierarchies using side information. To this end, a hierarchy is used as prior knowledge and words are generated in each hierarchy.

The main weakness of the work is that it relies upon a knowledge base. While WordNet is "relatively smaller" in size compared to knowledge bases such as Wikidata, where the model will struggle is when the knowledge bases are huge, e.g., several gigabytes in size. It will be good for the authors to mention which hierarchies the model can work and which it cannot.

While the authors have presented their detailed results, the question is why this work is important? The issue is that the model relies on a hierarchy that it supplied manually. There are already unsupervised topic models such as hLDA that can learn hierarchies, of course, not in the way that the authors have presented their work. However, it is still unclear how this work is a step-change in the topic modelling literature.

---

> ### Author Response · Authors · 2022-08-02
> **Response to Reviewer 1Lsj**
>
> Thank you for your time and the detailed feedback.
> The typos, more discussions about the real-world applications and the additional experiment have been addressed in the revised version, with the major changes marked in blue.
> Below we respond to your questions/criticisms.
>
> **w1**
>
> We follow the previous work [1] and use WordNet as the knowledge graph mainly because that WordNet provides 1) the brief definition of the concepts (or topics); 2) the hyponymy relation between two words, which is used to construct the pre-defined topic tree. Wikidata is a general knowledge base and it may not easy to obtain clean hyponymy relations between words. The node size in our topic tree is the number of all words plus the number of topics at each layer, e.g., $N = V + \sum_{l=1}^L K_l$. The vocabulary size $V$ is usually set as 2k-10k in topic models, and the topic number $K_l$ is usually set as 20-500, Thus the number of total nodes in most topic modeling tasks is less than 20k (the largest N in our experiment is 10,252). Empirically, the proposed model can handle most topic modeling tasks. It is hard to give an upper bound of the graph size.
>
> **w2**
>
>  Aiming at learning topic hierarchies and hierarchical document topical representation, hierarchical topic models have obtained increasingly research interest due to the promising performance. Most existing hierarchical topic models, such as hLDA and SawETM are designed only for the BoW assumption and trained in a pure data-driven paradigm by maximizing the likelihood of the observed data, which causes an over-concentration on high-frequency words. Moreover, the infrequent words might be assigned to irrelevant topics due to the lack of side information. The above two drawbacks could lead to human-unfriendly topics that fail to make sense to end users and limit their real-world applications. One of the main goals of this paper is to incorporate human knowledge into topic modeling to address the above shortcomings.
> We note that the introduced prior topic tree is often easy-to-obtain and contains human common sense about the relations between concepts and words. Besides, we also provide a script for automatic topic hierarchy construction with a certain heuristic rule according to the target corpora (Sec5.3 in the manuscript), and more details of the implementation can be found in our attached code. To summarize, the proposed model in this paper provides a novel knowledge-aware Bayesian deep topic modeling framework that incorporates domain knowledge into TMs and helps to improve the performance of hierarchical topic models. The proposed TopoicKG and TopicKGA give a strong baseline in the knowledge-based topic modeling literature.
>
> **Q1**
>
> As an unsupervised text analysis technique, the proposed model in the paper has the ability to discover a set of interpretable topics from a target corpora, which can be further used for corpora summary. For example, one can train TopicKG on all submitted papers at the NeurIPS2022 conference. The mined topics can be viewed as a brief summary of the hot research directions. Besides, one can also cluster the submitted papers according to their topic proportions, which is useful for Beginners and researchers in a certain field. What's more, TopoicKG allows users to provide their interesting topic structure to guide the learning of topic models. For example, a researcher has prior knowledge about the "Machine Learning " field, and he can provide his own "Machine Learning "  hierarchies (similar to Fig.1 in the manuscript) into TopicKG. As a result, the learned topic structure will meet with the provided structure and the document will also organize in such taxonomy.
>
> **Additional result on RCV2**
>
> We follow your advice and have added the result on RCV2 dataset at Fig.5 in the rebuttal revision. RCV2 contains 150,737 documents. From Fig.5 we can find that the proposed models have similar performance on 20NG and RCV2, which further demonstrates that the proposed models not only have a faster learning speed than TopicNet but also achieve better
> reconstruction performance on both small and large corpus.
>
> [1] TopicNet: Semantic Graph-Guided Topic Discovery. Duan et al. 2021.

---

> > ### Comment · Reviewer_1Lsj · 2022-08-08
> > **Post author response comment.**
> >
> > I thank the authors for their response. The authors have clarified most of my doubts. What I will request the authors to consider in the future is to ask the question with regards to how generalisable the model is. For instance, I do not see that just using a specific hierarchy can help when there is already a noisy large-scale knowledge base such as Wikidata. In the current form, I see that this work might not be very useful to many researchers.

---

> > > ### Author Response · Authors · 2022-08-08
> > > **Author Response**
> > >
> > > We thank you for the further discussion and valuable comments.
> > >
> > > We agree with you that it is important to incorporate large-scale knowledge base into Bayesian generative models. Human common sense in general knowledge base can be viewed as the side information that helps to improve such pure data-driven models. Our proposed TopicKG provides an alternative to jointly model the observed document and the specific hierarchy in a Bayesian manner for topic modeling. Note that the topic hierarchy can be viewed as a special sub-graph of a large graph, and there may be some potential research directions  that can generalize our model to many tasks.

---

### Official Review · Reviewer_o1DX · 2022-07-12

**Rating:** 4
**Confidence:** 4
**Soundness:** 3 good
**Presentation:** 2 fair
**Contribution:** 2 fair

**Summary:**

This paper proposes an improved embedded topic model that considers an existing knowledge about the topic hierarchy that can potentially alleviate biased and long-tail topics and words. It considers such hierarchy information as a matrix representing the interactions of topics following a Bernoulli distribution. The experiments include various aspects of the approach. They use four datasets and compare the proposed approach with five baselines. Also, the interpretability and the coherence of the topic is empirically shown and examples are provided.

**Questions:**

I appreciate the authors for the efforts to incorporate prior knowledge to topic modeling. While there is some evidence of the improved performance, it was not sufficiently explained in general. Please refer to W1-W3 to address these questions. Especially, deep diving why this modeling is better than the baselines is very important to understand where the strength is coming from. Also, clarifying what comes from an existing approach, and what is novel can improve the readability of the paper greatly, and that way, this paper doesn't need to try to explain everything.

Additional comments/questions follow.
- Q1. A GCN was used to consider the structure of the topic tree and update the node weights. How was the neural network training and the Bayesian model optimization done together?

- Typos: EBLO -> ELBO. consists -> consists of. widely-used -> widely used. two adjacent layer -> two adjacent layers.

**Limitations:**

I don't see a negative societal impact of this work.

**Strengths And Weaknesses:**

Strengths.
- S1: The proposed approach provides a way to incorporate the topic hierarchy from an external knowledge source.
- S2: The approach was extensively evaluated with five baselines and four datasets.
- S3: The examples and interpretability of the derived topics and their hierarchy are provided to understand the outcome of the approach.

Weaknesses.
- W1: The presentation can be improved. There is no overview of the approach to explain the components, and a few components and concepts appear without much prior context. For example, "encoder" appears without where it is exactly being used. Same for "topic aggregation". "count vector" was used only once without definition.
- W2: This paper does not provide the related work description for the existing ETM this work is based on in the related work section. It also doesn't distinguish that in the method section. Thus, it is hard to evaluate the novel contribution of this paper compared to the existing approach. For example, much of the framework comes from the SawTooth paper, but this paper failed to include or summarize the overall structure.
- W3: Many parts of the paper are not justified or explained enough. For example, how modeling the relation with a Bernoulli distribution complement Phi modeling the relations between layers in SawTooth, or why the proposed approach is better was not explained. How is the adaptive structure different from just fine-tuning the parameters? The paper quickly attributes the improved performance to the joint tree likelihood, but how exactly is it better than the other approaches such as SawTooth or TopicNet? What information does it capture that SawTooth and TopicNet do not?

---

> ### Author Response · Authors · 2022-08-02
> **Response to Reviewer o1DX (1/2)**
>
> Thank you for your time and the detailed feedback.
> The typos, some undefined concepts, more related description about the existing ETMs and more explanations about the improvement have been addressed in the revised version, with the major changes marked in blue.
> Below we respond to your questions/criticisms.
>
> **w1**
>
> Thank you for helping us improve the representation.
> we have
> **1)** provided a brief overview of TopicKG at Fig.2 in the submitted manuscript, which contains a Weibull upward-downward variational encoder and the GCN-based topic aggregation module. We will follow your advice and replace Fig.2 with a more detailed one in the final revision.
>  **2)** revised the title of Sec3.3 "encoder" to "Inference network of TopicKG";
> **3)** revised the "count vector" to "count-value vector", which means a vector whose elements are non-negative integers;
> **4)** added a detailed description of the "topic aggregation module", e.g., the topic aggregation module updates the node embedding by aggregating the neighbor information via the GCN layer.
>
> **w2**
>
> We follow your advice and have added more descriptions and discussions for ETMs in the related work (Sec2 in the revision).
> The proposed ToipcKG and TopicKGA are built upon ETM where words and topics are regarded as the embedding vectors, but different from most existing ETMs that are designed only for the BoW assumption and trained in a pure data-driven paradigm, which may fail to learn high-quality topics when a large number of topics and their relations need to be modeled in the deep topic modeling tasks [1,2], our proposed TopicKG provides a new knowledge-aware Bayesian deep topic modeling framework that jointly models the BoW representation and the pre-specific topic tree (knowledge graph) via the shared embeddings vectors. This allows ETMs (such as SawTooth) have the ability to make use of the prior knowledge as the side-information to guide the learning of topics and improve their performance.
>
> Although both SawETM and TopicKG employ the gamma believe network to model the generative process of the document likelihood $\mathbf x$:
> \begin{equation}
>      \mathbf x_j^{(l)} \sim \text{Gamma}(\mathbf \Phi^{(l+1)} \mathbf \theta_j^{(l+1)}), \quad with \quad l=1,...,L-1  \quad \mathbf \Phi^{(l)} = \text{Softmax}({\mathbf e^{(l-1)}}^T \mathbf e^{(l)}) \quad \quad
>     \mathbf x_j \sim \text{Pois}(\mathbf \Phi^{(1)} \mathbf \theta_j^{(1)})
> \end{equation}
> they have a distinct difference in the learning of topics. SawETM views the topic embeddings $\mathbf e^{(l)}$ at l-th layer as the trainable vector that  is directly  learned by the ELBO of $\mathbf x$. To incorporate prior knowledge into topic modeling, TopicKG first extracts two binary knowledge matrices from WordNet: topic-topic relationship matrix $\mathbf S$ and the topic-word relationship matrix $\mathbf C$. The generative model of those two knowledge matrix at layer l in TopicKG are formulated as:
>
> \begin{equation}
>      S_{k_1, k_2}^{(l)} \sim {Bern}(\sigma({\mathbf e_{k_1}^{(l-1)}}^T \mathbf W {\mathbf e_{k_2}^{(l)}})), \quad \quad
> C_{v k}^{(l)} \sim {Bern}(\sigma({\mathbf e_v^{(0)}}^T {\mathbf e_k^{(l)}})),
> \end{equation}
> Thus the topics at each layer are learned from both the likelihood of $\mathbf x$ and the prior knowledge. In other words, TopicKG incorporates domain knowledge as the side-information to guide the learning of topics, which is ignored in most existing ETMs.

---

> > ### Author Response · Authors · 2022-08-02
> > **Response to Reviewer o1DX (2/2)**
> >
> > **w3**
> >
> > We apologize for not enough explanation. Bellow let us provide more details to address your main concerns. As discussed in w2, two binary knowledge matrices are introduced as the side-information to improve ETMs. Thus the proposed TopicKG models such binary observation under the Bernoulli likelihood, which provides a theoretical guarantee that two nodes have close semantic distance at the embedding space if there is a link between that two nodes. For example, for the directed binary matrix $\mathbf S$ (we here omit the superscript), $S_{k_1,k_2}=1$ means there is a link from topic $k_1$ to topic $k_2$ (topic $k_2$ is a child node of topic $k_1$). Thus the Bernoulli likelihood in TopicKG will guide the learning of corresponding topic embeddings that have enough semantic similarity to make sure ${\mathbf e_{k_1}}^T \mathbf W \mathbf e_{k_2}$ has a large value. The topic-too-word matrix $\mathbf C$ has similar findings, $C_{vk}=1$ will guide the k-th topic embedding and v-th word embedding have enough semantic similarity. As a result, both the introduced $\mathbf S$ and $\mathbf C$ will guide the learning of word and topic embeddings via the provided domain knowledge and therefore guide the $\mathbf \Phi$ modeling in SawETM.
> >
> > To address the mismatch issue between the pre-defined topic tree and the target corpora, TopicKG is further extended to TopicKGA, which allows the prior topic structure can be finetuned via the graph adaptive technique, resulting in better document representation. The core idea behind the TopicKGA is the gradually adaptive topic structure in the annealed training algorithm (Eq.8 in the manuscript). That is to say, the ground-truth topic structure has the ability to add new knowledge (e.g., new nodes and new edges) and reinforce itself according to the current corpora, resulting in an increasingly matching topic tree as the training progresses. while the fine-tuning only updates the word and topic embeddings, it does not change the structure of the topic tree. Fine-tuning fails to add new knowledge according to the current corpora.
> >
> > We attribute the improvements in our proposed models to 1), domain knowledge incorporation. Different from most existing ETMs (e.g., SawETM) that learn topics only from the word co-occurrence patterns in the BoW vector, our proposed TopicKG incorporates prior domain knowledge into ETMs by jointly modeling the document and the knowledge matrix in the Bayesian framework. We note that such prior knowledge can be regarded as the side-information to guide the learning of word and topic embeddings, resulting in high-quality topic hierarchies; 2), adaptive topic structure. Unlike previous knowledge-based topic models (e.g., TopicNet) that use the fixed topic tree in the whole training process and ignore the mismatch issue between the pre-defined topic tree and the target corpora, which could introduce the noise and confuse the model during the learning. To this end, our TopicKGA allows the pre-defined topic tree can be fine-tuned according to the current corpora via the graph adaptive technique, resulting in better document representation.
> >
> > **Q1**
> >
> > To jointly train all the parameters in the GCN layer and the Bayesian model end-to-end, we derive an efficient algorithm for approximating the posterior of topic proportion with amortized variational inference and view the word and topic embeddings as the deterministic learnable vectors, resulting in the final objective function, as formulated in Eq.6 in the manuscript. All parameters can be optimized via the gradient descent algorithm. We refer reviewers to the appendix for more training details.

---

> ### Author Response · Authors · 2022-08-09
> **Desire for further discussion**
>
> Dear Reviewer o1DX:
>
> We thank you again for the valuable comments and for helping us to improve our revision. In the past few days, we tried our best to address your questions and concerns and revised the paper following your suggestions, including more related work description for NTMs, clearer definition of several concepts, highlighting the main contributions, explaining where the improvement is coming from, etc.
>
> We appreciate it if you could let us know whether our responses and revisions are able to address your concerns, We are willing to have a discussion with you and would be happy to address any further concerns.
>
> Best regards,
>
> Paper8773 Authors

---

### Author Response · Authors · 2022-08-08
**Feedback from reviewers**

Dear Reviewers,

We appreciate it if you could let us know whether our responses and revisions are able to address your concerns. We're happy to address any further concerns.
Thank you,

Best wishes.

Paper8773 Authors

---

### Meta-Review · Area_Chair_N5ip · 2022-08-30

**Recommendation:** Accept
**Confidence:** Certain

**Metareview:**

Learning hierarchies of themes has been a challenging field that saw considerable work about ten years ago but has largely been abandoned due to poor results and inefficient algorithms. I'm interested to see this and other work reviving interest. My prior for hierarchical topics is that they are difficult to construct because the search space is large and doing a good job requires prioritization and abstraction at a high level of AI. I like the semi-supervised approach here as a heuristic to guide the construction of hierarchies. WordNet is often difficult to apply in practice, so I share the reviewers interest in alternatives that would scale.

Reviewer complaints largely centered around clarity. The authors provided extensive responses which the reviewers found compelling and satisfactory. There is generally solid to strong support for the paper.

**Award:**

No

---

### Decision · Program_Chairs · 2022-09-14

Accept